# Botanical Assessment of Disturbed Urban Population of Threatened Gopher Tortoise (*Gopherus polyphemus*) Habitat in SE Florida During Drought

**DOI:** 10.3390/biology14081038

**Published:** 2025-08-12

**Authors:** George Rogers

**Affiliations:** Palm Beach Gardens, Palm Beach State College, 395 Mallard Pt., Jupiter, FL 33458, USA; rogersg515@gmail.com

**Keywords:** habitat fragmentation, gopher tortoise habitat, grazing, *Richardia grandiflora*, *Paspalum setaceum*, *Fimbristylis cymosa*

## Abstract

Gopher tortoises are a threatened species in South Florida. They face several perils, including habitat loss and degradation, diseases and parasites, crowding, and disturbances. The present study monitors the main weedy foodplants of a precarious urban population in Palm Beach County, Florida through the severe drought of late winter and spring 2025. The primary finding was that the main foodplants, including an exotic weed (Mexican-Clover, Richardia grandiflora), showed substantial resilience to drought and to drought coupled with intense grazing. A different nonnative species avoided by the tortoises (Hurricane-Grass, Fimbristylis cymosa) appears to be expanding coverage as tortoises graze its competitors.

## 1. Introduction

Gopher tortoises (Figure 1A) are indigenous to the Southeastern U.S. Coastal Plain, burrowing mostly in dry sandy habitats [1]. Sharing its preference for well-drained uplands with urban development, the species has a history of overlap with human activities, prompting conservation issues. This is particularly true in Southeast Florida, which is naturally largely wetland unsuitable for tortoises and for urban development. Dry habitats are restricted to a thin coastal strip shared with highways, RR tracks, and decades of residential and tourism development. According to the National Oceanic and Atmospheric Administration Southeast Regional Land Cover Change Report, 1996–2010 (https://coast.noaa.gov/data/digitalcoast/pdf/landcover-report-southeast.pdf (accessed 6 August 2025), “With a loss of 3563 square miles [South Carolina through Florida], upland forest was the land cover with the greatest net decrease in area”. Problems include habitat fragmentation and degradation, as well as crowding, forced relocations with unnatural population mixing, disease and parasites, burrow interference (Figure 1B), and predation mostly of young and eggs [2,3,4,5,6,7]. The State of Florida lists gopher tortoises as a threatened species [1] Southeastern Florida is overrun with urban sprawl from Miami northward along the Atlantic Coast. As noted by the FFWC [1] p. 14, referring to the tortoises, “their current range in South Florida is limited by unsuitable habitat and urbanized areas”.

Plant-centered gopher tortoise research has largely featured dietary inventory [8,9,10] or fecal dispersal of plant propagules [11,12]. Lloyd et al. [13] in Alabama found tortoise exclusion vs. open grazing to increase plant cover and to diminish species richness and diversity. Richardson and Stiling [14] likewise found tortoise exclusion to diminish species richness and diversity, and to increase dominance by *Heliotropium polyphyllum* (now *Euploca polyphylla* (Lehm.) I. M. Melo & Semir, Boraginaceae). Gopher tortoises have broad vegetarian dietary habits [8,15,16] with situational preferences [9,10]. How, or if, drought coupled with year-round additional stresses in South Florida limits forage availability remains poorly known. Year-round stresses include nutrient-poor soils, extreme heat, anthropogenic disturbance including dogs, unnatural vegetation, aggressive management in places, and inconsistent weather.

Being conspicuous and charismatic, gopher tortoises garner attention ranging from popular media and educational materials to zoological research. The Gopher Tortoise Council [17] maintains an online bibliography. Despite considerable disparate attention, ever-changing urban challenges create gaps in practical knowledge. Whitfield et al. [6] surveyed the tortoise populations in Palm Beach County, the area of the present study, and southward, and urged continued research in this region. As they related, the Southeast Florida tortoise populations are so limited that many geographically broad studies have more or less ignored them. The southeast Florida tortoises find themselves in site-specific, human-modified habitat islands, each an “experiment” with its own stressors. Biologists at Florida Atlantic University in nearby Broward County recently reported a tortoise die-off of uncertain causes at the university’s ecological preserve [18].

Hairston et al. [19] generalized that plants are seldom depleted by herbivores except under unnatural crowding. With an eye to such forced crowding in a floristically depauperate weedy urban habitat fragment, the present study is a botanical “stress test” of the main grazed groundcover species during drought.

As of 6 May 2025, the U.S. Drought Monitor [20] placed all of Palm Beach County under “severe” and “extreme drought”, this reflecting an increase from ca. 50% “severe drought” since 2/11/25 (Figure 2A). Multiyear average rainfall in Palm Beach County [21] during the dry season is January 80 mm (vs. 26 mm in 2025), Feb 72 mm (vs. 25 mm), March 117 mm (vs. 48 mm), April 93 mm (vs. 14 mm), May 115 mm (vs. 123 mm mostly in one event), and June 211 mm (106 mm). After ca. mid-May 2025, the drought abated gradually with intermittent increasing rainfall to transition into the summer wetter season [22] (Figure 2B). (Although in the summer “wetter” season extending past the end of the study, June 2025 continued a dry trend, having only about half of its average rainfall.) The grazed vegetation was monitored through late May and into June.

The research questions (RQ) are as follows:

RQ 1. Did grazed areas have different species compositions than otherwise similar non-grazed areas?

RQ 2. Under tortoise-free, and under grazed conditions, did the main groundcover foodplant species increase, decrease, or remain stable through the drought?

RQ 3. What impact, if any, did the tortoises have on the abundant yet dietarily disfavored nonnative sedge (*Fimbristylis cymosa* R.Br., “Hurricane-Grass”, Cyperaceae)?

## 2. Materials and Methods

### 2.1. Study Site, Site Conditions, and Tortoise Population

Geographic points mentioned in the text are listed with grid coordinates along with calendar dates in the Appendix A. The study site (Figure 3 and Figure 4) straddles uninterrupted two properties hosting over 130 gopher tortoise burrows as well as numerous abandoned burrows. The northern portion is a narrow RR trackside and power line extension of the much larger “Botanica Preserve”, a protected wetland irrelevant to the present study. The southern portion of the study site is a berm extending southward between the RR track and a drainage canal. The overall site is T-shaped and fully encircled by roads, residential development, RR tracks, drainage canal, and golf course. The full east border is a RR track separated from tortoise habitat by physical barriers and a tangle of weedy growth. Near the central eastern border, the “commons” (Figure 5A) is the point of highest tortoise concentration, surrounded by burrows and with as many as six individuals grazing there simultaneously. The commons (80 m × 20 m) is open, sparsely covered with a mix of bare sand and the mat-forming weed *Richardia grandiflora* (Cham. & Schltdl.) Steud. (Mexican-Clover, Rubiaceae) plus less-abundant intermixed species. The Florida Invasive Species Council [23] designates *R. grandiflora* as a Category II invasive exotic. Extending north from the commons, a paved road (north path, Figure 5B) is flanked by tortoise burrows as it follows the RR and power line about 484 m to the north gate opening onto an automobile road. Housing no tortoises, the “north extension” leads northward as rough roadside from the north gate.

Partly paved and partly weedy sand, and bordered with tortoise burrows, the south path runs from the commons southward approx. 258 m to the freely transited and circumvented south gate. (This is the only gate extending the tortoise population into continued suitable habitat.) From the south gate, the south extension (Figure 4 and Figure 5C) continues many km southward on a spoils berm between the RR and a steep-sided drainage canal. The south extension has unevenly distributed tortoise burrows, with an active cluster near the south gate, then southward with few active burrows for 560 m to a burrow cluster referred to here as the “pipe” (a canal pipe-crossing, the southernmost point monitored). South of the pipe, I am aware of no additional burrows, having explored further southward to a major crossroad 1600 m south of the commons and beyond the scope of the study.

The west path (Figure 5D) is a segment of a paved maintenance road serving an electrical power line. The west path extends from the commons through open pine woods and *Fimbristylis* lawn (Figure 5E) 237 m to the west gate, this opening onto tortoise-impassable automobile roads and mowed turf. Across those barriers, the power line and maintenance road continue westward as the west extension with scattered tortoise burrows and tortoise-free areas.

The study area has four chief coverage types:

A. Road, mostly paved or weedy sand. The uncurbed paved roads within the site are separated from street traffic, and are traversed by tortoises, pedestrians, bicycles, and motorized maintenance vehicles.

B. Sandy sparsely vegetated open spaces, mostly the commons and roadsides. The dominant four groundcover species on these open spaces in order of prevalence were *Richardia grandiflora*, native *Paspalum setaceum* Michx. (Thin Paspalum, Poaceae), native *Phyla nodiflora* (L.) Greene (Fogfruit, Verbenaceae), nonnative *Fimbristylis cymosa* (especially in the *Fimbristylis* lawn, Figure 5E), and nonnative *Borreria verticillata* (L.) G. Mey. (False Buttonweed, Rubiaceae). This coverage type is the grazed area. Note that the vegetation tends to be sparse, usually mixed with exposed sand (Figure 5A), and dominated by weedy *R. grandiflora* (Figure 5F) and *P. setaceum*. The first three species are strongly grazed (Appendix A).

C. Pine woods with slash pine (*P. elliottii* Engelm., Pinaceae.) and scrub pine (*Pinus clausa* (Chapm. ex Engelm.) Vasey ex Sarg.). Most of the understory is saw palmetto (*Serenoa repens* (W. Bartram) Small, Arecaceae). Tortoises were rare in the wooded areas, then generally in sandy openings under pines. Most burrows are at the borders between the woods and open ground.

D. Vine tangles and tall weeds.

The tortoises are crowded. Including the south extension, with 136 active burrows counted site-wide. Whitfield et al.’s [6] 50% burrow occupancy estimate gave a present project population of ca. 68 tortoises on 7.7 hectares, thus a density of about 8.8 tortoises/hectare. Because tortoises are rare in the wooded areas except as noted above, the effective density is close to 68 tortoises on 2.4 ha favored habitat. The highest density of eight sites Whitfield et al. [6] reported was roughly five tortoises/ha. Ceballos and Goessling [9] tallied 300 tortoises on 99 hectares (3/ha). The Florida Fish and Wildlife Commission [1] management guidelines give a maximum of approx. 5–10/ha under favorable conditions.

Although a formal tortoise population assessment is beyond the scope of this botanical project, informal observations, mostly Feb–early June 2025, may be of interest: tortoises of all sizes, including hatchlings, were present. The site is heavily visited by people, often with dogs on and off the leash. Dog fecal deposits were eaten by the coprophagous tortoises (Appendix A). Urbanized coyotes (*Canis latrans*) occur in the general area, including, anecdotally, the present site. Additional commonplace urban mammals, including raccoons (*Procyon lotor*), rabbits (*Sylvilagus floridanus),* and opossums (*Didelphis virginiana*) are present. Natural tortoise immigration is unlikely because the site is completely surrounded by barriers and unsuitable habitat. However unofficial relocations of tortoises from golf courses, construction zones, and roadsides are likely but of unknown frequency. Emigration is likewise mostly blocked except for limited expansion along the south extension. During the study period, I encountered on site the remains of seven dead tortoises, two of them immature.

Additional grazers are present, mainly rabbits. Present with unknown grazing impact are invasive green iguanas (*Iguana iguana,*
Figure 1B). Truglio et al. [24] described green iguana interference with gopher tortoise burrows.

The dominant groundcover in the open areas was “*Richardia*-carpet” (Figure 5F) interspersed with open sand. To define “*Richardia*-carpet”, the groundcover consists of clumps of prostrate *R. grandiflora* acting as a matrix for *Paspalum setaceum* and less abundant low species (Figure 6). In some limited areas, *Paspalum* was the main groundcover. The substrate surface becomes intensely hot (Figure 7A).

Site management is minimal. To my knowledge, there was no mowing during the study period. Two exclosures were lost to vandalism. All exclosures, except the southernmost at the pipe, were removed to end the study period on 16 June 2025 due to on-site roadside vegetation clearing corresponding with end-of-drought season regrowth.

### 2.2. Vegetation Quadrat Monitoring Through Drought Period

A summary of the monitoring setups is in Appendix B Table A1. Beginning and data check dates during the drought are in the figure captions.

#### 2.2.1. Grid Quadrats and Floristic Composition

To determine the floristic compositions of *Richardia*-carpets with vs. without tortoises, square wire mesh quadrats having 64 five cm grid cells (Home Depot, Atlanta, GA, USA) were applied to 19 scattered vegetation patches as detailed in Appendix B Table A1. The patches were selected for edaphic similarity to each other, for dominance by carpet vegetation, for freedom from shade, for level dry ground without puddling, for unambiguous tortoises presence vs. absence, for minimal road-and forest-edge effects, and for freedom from other interferences. The starting point for quadrat placement was chosen by over-shoulder quadrat tosses. At each patch, the quadrats were placed 10 times parallel to pathways. The same pattern was used at each patch: corner to corner in a line parallel to the road, the quadrat edges aligned and alternating left and right along a common axis. Data were recorded as the dominant species in each of the 25 cm^2^ grid cells, 640 cells/patch, 200 quadrat placements.

#### 2.2.2. Richardia-Carpet Exclosures to Prevent Grazing Through Drought

To assess drought effects on the *Richardia*-carpet with and without grazing, 14 exclosures were placed in grazed areas, as detailed in Appendix B Table A1. Two of the exclosures were fences 60 cm × 60 cm near the south gate. The other exclosures were black wire cages 31.5 cm × 22 cm (Greenbrier Intl., Chesapeake, VA, USA) chosen to minimize passerby attention and to match the standing-position camera-aspect of a Canon T6S digital camera with Canon EFS 55–250 mm lens used to record vegetation coverage. Criteria for exclosure placement, in addition to minimizing attention, were open grazing areas without impeding burrow entrances. Quadrats of the same size served to record non-exclosed neighboring grazed quadrat sites. (Quadrats other than the wire mesh grid cells noted above made by the author using 12.7 mm diam. PVC plumbing pipe.) Their centers were marked using a tagged nail into the soil to allow exact repeated quadrat placement during photography.

Because the commons area was particularly heavily grazed and disturbed by tortoise movements, the *Richardia*-carpet there was given special attention. That entailed using five 60 × 40 marked quadrat positions determined by over-the-shoulder quadrat tosses. For all open grazing quadrats, quadrat centers were marked, and the frames placed only during photography. Exclosures were removed and momentarily replaced with the quadrat frame for photography. Data were collected for all *Richardia*-carpet exclosure and quadrat sites photographically in the same fashion except for using a Canon 28–80 mm wide-angle lens for the larger quadrats in the commons. Photos were taken vertically downward from a standing position using the quadrat edges to define photo margins. The resulting photographs were analyzed using ImageJ software (1.54p 17 Feb 2025) for colorimetric measurement of the fraction of each quadrat covered by *Richardia*-carpet.

#### 2.2.3. Paspalum Photo Measurements to Assess Clump Expansion or Contraction During Drought

The “3030” comparisons were between a grazed locale (pipe) and an otherwise similar tortoise-free spot (“albizias”), both along the south extension.

For both spots, an east–west line was marked from the RR edge to the canal edge. Proceeding northward from those lines along the berm, the first 30 *Paspalum* rosettes encountered at each spot were photographed in a 15 cm × 15 cm quadrat frame. The fraction of the frame occupied by its enclosed *Paspalum* rosette was determined using the area measurement capability in Photoshop (26.7) to measure the proportion of the quadrat frame occupied by the rosette. Such measurements were repeated from the same lines at the same spots periodically through the drought.

As a redundant check on the above “3030” substudy, a repeat “transect” substudy used two transects along the south path, both in grazed areas (pipe, south gate), to record the transect positions of 10 rosettes at each site; the *Paspalum* coverage measured in the same fashion as above.

#### 2.2.4. Fimbristylis Exclosures to Exclude Grazing During and Shortly After Drought

Six 22 cm × 31.5 cm black wire cage exclosures as described above were placed in the *Fimbristylis cymosa* lawn, the placement criteria being strong uniform *Fimbristylis* coverage, minimal passerby visibility, and avoidance of tortoise burrows. During the drought, the *Fimbristylis* rosettes became mostly dormant and brown, remaining green at the basal centers. During this period, tortoises roamed the *Fimbristylis* lawn selecting other green species from among the *Fimbristylis* rosettes. During most of the drought, vegetation outside the exclosures was visually indistinguishable from that inside. However, at the end of May and early June, grass expanded within the exclosures. For this study, therefore, monitoring was extended into June. Grass (not *Fimbristylis*) coverage was assessed as for *Paspalum* but using the 22 × 31.5 quadrat.

### 2.3. Graphics and Statistics

Graphs and calculations used R 4.5.1 with the ggplot2 extension (https://www.r-project.org/). Because the research questions concerned tendencies rather than testing hypotheses, the statistical analysis was Bayesian (Appendix C Figure A1), using R package MCMCpack 1.7-1 applying Monte Carlo Markov Chain sampling with 10K iterations. Priors were based on the small extra studies: the “commons” Appendix A) for *Richardia*-carpet exclosures, and the “tape transect” data (Appendix A) for *Paspalum* rosettes. Both of those studies concerned only grazed conditions. Priors for the beginnings of the non-grazed conditions used variances and mu values from the grazed priors since the non-grazed studies started from grazed conditions. For the post-drought in the grazed studies, the prior mu values were projected to be the starting value plus 10%.

## 3. Results

Results are summarized in Table 1. Plant species the tortoises particularly avoided were nonnative *Catharanthus roseus* (L.) G. Don (Madagascar-Periwinkle, Apocynaceae), *Physalis walteri* Nutt. (Walter’s Groundcherry, Solanaceae) and as discussed further below, *Fimbristylis cymosa*. Although some fleshy-fruit-bearing species border the roads, none were bearing fruit during the drought.

With respect to the research questions:

RQ1. Did tortoise grazing influence overall herbaceous species compositions? The grid quadrat results (Figure 6) show stronger representations of *Paspalum setaceum* and of *Phyla nodiflora* when tortoise-free. *Fimbristylis cymosa*, apparently facilitated by grazing, is discussed separately below. *Richardia grandiflora* showed little difference with respect to grazing.

RQ2. In tortoise-free exclosures, did the ungrazed *Richardia*-carpet species decrease, increase, or remain stable? *Richardia*-carpet increased a net ca. 27 percent through the drought (Table 1, Figure 7B brown lines. See similar Bayesian results in Appendix C Figure A1). In the case of ungrazed *Paspalum*, with tortoises (but not rabbits) absent, at the 3030 sites, the *Paspalum* coverage decreased ca. 17.1 percent (Table 1, Figure 7D brown lines). The Bayesian results likewise show a minor decrease (Appendix C Figure A1).

Grazed *Richardia*-carpet exclosure sites (Figure 7B blue lines) showed a coverage decline of ca. 7.3 percent, echoed in the Bayesian outcome (Appendix C Figure A1). The separately monitored grazed *Richardia*-carpet in the commons declined similarly, 23.6% during the study (Table 1, Figure 7C). In the case of the *Paspalum* 3030-rosette substudy (Figure 7D blue lines), there was a ca. 38% increase with grazing. The transect-based check on grazed *Paspalum* (Figure 7E), conflictingly, gave a ca. 29.3% decrease over essentially the same dates (see Discussion below).

RQ. 3. What impact, if any, did the tortoises have on the *Fimbristylis cymosa* lawn? Late in the study as drought abated, the *Fimbristylis* regreened quickly, and grasses rapidly became increasingly prevalent in the exclosures, increasing 30.6% relative to the earlier date (Table 1, Figure 7F). Note the large difference between grazed and ungrazed conditions for both dates.

## 4. Discussion

The general outcome revealed substantial groundcover robustness to drought, and to drought combined with grazing. *Richardia*-carpet increased despite drought when ungrazed and declined only slightly when grazed, even in the heavily grazed groundcover-depauperate commons. *Paspalum setaceum* showed comparable high resilience, with little overall effect of drought or of drought with grazing. The conflicting beginning and endpoint for that species grazed in Table 1 for 3030 (increase) vs. transects (decrease) diminishes in impact when viewing the fundamental similarities of the all-season patterns (Figure 7D blue bars and Figure 7E), with a small final uptick in Figure 7D. The Bayesian outcome (Appendix C Figure A1) emphasizes that the combined stresses had little impact on *P. setaceum*.

*Richardia grandiflora* is prevalent in South Florida and far beyond in coarse, disturbed, exposed, nutrient-poor sandy habitats, displaying durable prostrate growth, rooting at the nodes (Figure 8B). MacDonald and Mushinsky [8] listed this species as consumed abundantly by tortoises. Figueroa et al. [25] found it the fourth-most abundant seed in gopher tortoise scats in the Miami Rocklands. Stilson [26] listed *Richardia* among the favored genera by juvenile tortoises. Moura et al. [27] discussed *R. grandiflora* adaptations allowing its exceptional tolerance of high temperatures, intense exposure, poor dry sandy soils, and wind. *Paspalum setaceum* is often encountered on abused, dry, sterile substrates. It tillers from rosette clumps freely, even after grazing in drought (Figure 8C). The groundcover carpet was not uniform across the study area, being most depleted in the most crowded spots, and enhanced where water and nutrients collect, such as depressions and adjacent to pavement (Figure 8D).

Foreshadowing the present study, Richardson [28] and Richardson and Stiling [14] listed *Fimbristylis cymosa* as “not preferred by tortoises” and as “avoided”. MacDonald and Mushinsky [8] likewise found *F. cymosa* unpreferred. Although the tortoises or rabbits may remove or damage the protruding flower scapes, on no occasion did I see a tortoise consume *F. cymosa* foliage. However, tortoises wandered through the *Fimbristylis*-lawn along the west path picking out interspersed other greens (Appendix A). As the season changed and drought abated, grasses expanding in the *Fimbristylis*-lawn exclosures (Figure 7F) suggested that grassy competitor removal facilitates the *Fimbristylis* dominance. With a similar prior observation, Richardson and Stilling [14] mentioned *F. cymosa* being outcompeted within gopher tortoise exclosures. (See [29] for *F. cymosa* nonindigenous status in Florida.)

Although observed infrequently, tortoises supplemented their typical open-area grazing with forest-edge vines (*Smilax auriculata* Walter (Greenbriar, Smilacaceae), *Cissus verticillata* (L.) Nicolson & C. E. Jarvis (Possum-Grape, Vitaceae), *Vitis rotundifolia* Michx. (Muscadine Grape, Vitaceae), *Parthenocissus quinquefolia* (L.) Planch. (Virginia Creeper, Vitaceae) (Appendix A). Tortoises occasionally consumed additional species, including *Ambrosia artemisiifolia* L. (Ragweed, Asteraceae, Figure 8E), seedling *Lepidium virginicum* L. (Pepper-Grass, Brassicaceae), and *Chamaecrista fasciculata* (Michx.) Greene (Partridge-Pea, Fabaceae).

## 5. Conclusions

The main finding was that a Category II exotic invasive weed (*R. grandiflora*) and a native but weedy grass (*P. setaceum*) appear to be resilient tortoise fodder in disturbed and vegetatively depauperate urban populations. A secondary finding was that a third weedy, non-native “unpreferred” and drought-dormant weed (*F. cymosa*) appeared to be capable of expanding as tortoises selectively remove grassy competitors.

The present study, however, examined merely one drought across one idiosyncratic site. At a different time and/or place, the balance between groundcover depletion and robustness may tilt differently. The stresses from drought and grazing, as well as from sterile soils, extreme sun, surface drying, inedible weeds, physical damage (vehicles, foot traffic, tortoise ground-scraping), and more are of unknown relative importance. Further monitoring may reveal if groundcover depletion immediately around crowded burrow clusters prompts relocation. This is of special interest in a confined and crowded habitat patch. A hint of such suboptimal relocations are burrows into the margins of the railroad bed (Figure 8F). Tortoise interference by invasive green iguanas needs further study as well.

The tortoises are coprophagous, consuming tortoise scats as well as those from the abundant dogs (Appendix A). As the drought diminished, the amount of uneaten tortoise scat appeared to increase, although was not quantified. It is unknown to what extent tortoise and associated animal waste may influence the plant cover, nor are the dietary risks of dog feces on gopher tortoises known, if any.

In a minimally managed site, such as the present, even low-cost, easy adjustments may be helpful. Although *Richardia grandiflora*, being an invasive exotic species, may be generally disfavored by conservationists, situational efforts to encourage it on urban tortoise habitats, perhaps merely shallow puddle depressions, may help tortoises more or less dependent on that surprisingly durable species as well *P. setaceum* and others. Additionally, that tortoises during drought (and likely at other times) eat forest-edge vines growing outside of the usual grazing areas suggests letting vines persist within reach during pathway maintenance in tortoise habitat.

## Figures and Tables

**Figure 1 biology-14-01038-f001:**
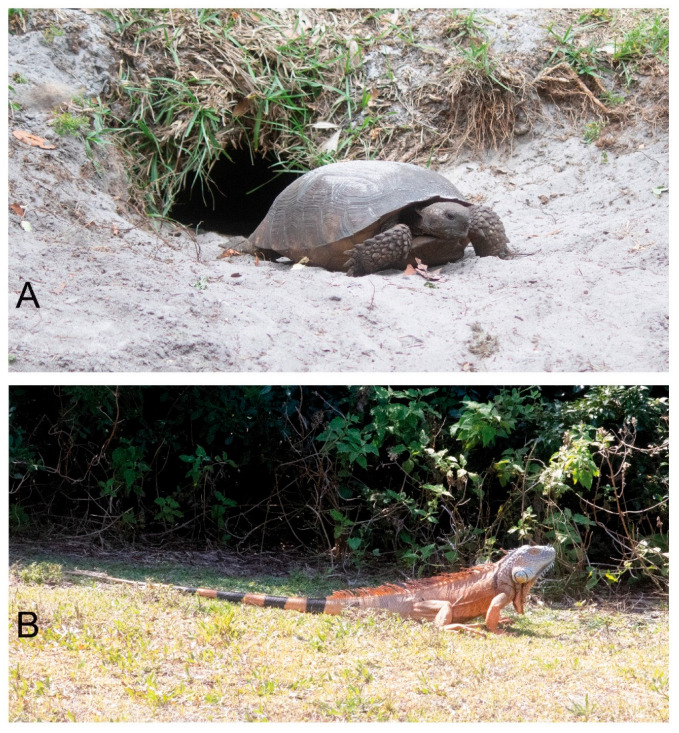
Large reptiles on site. (**A**) Gopher tortoise. (**B**) Green iguana.

**Figure 2 biology-14-01038-f002:**
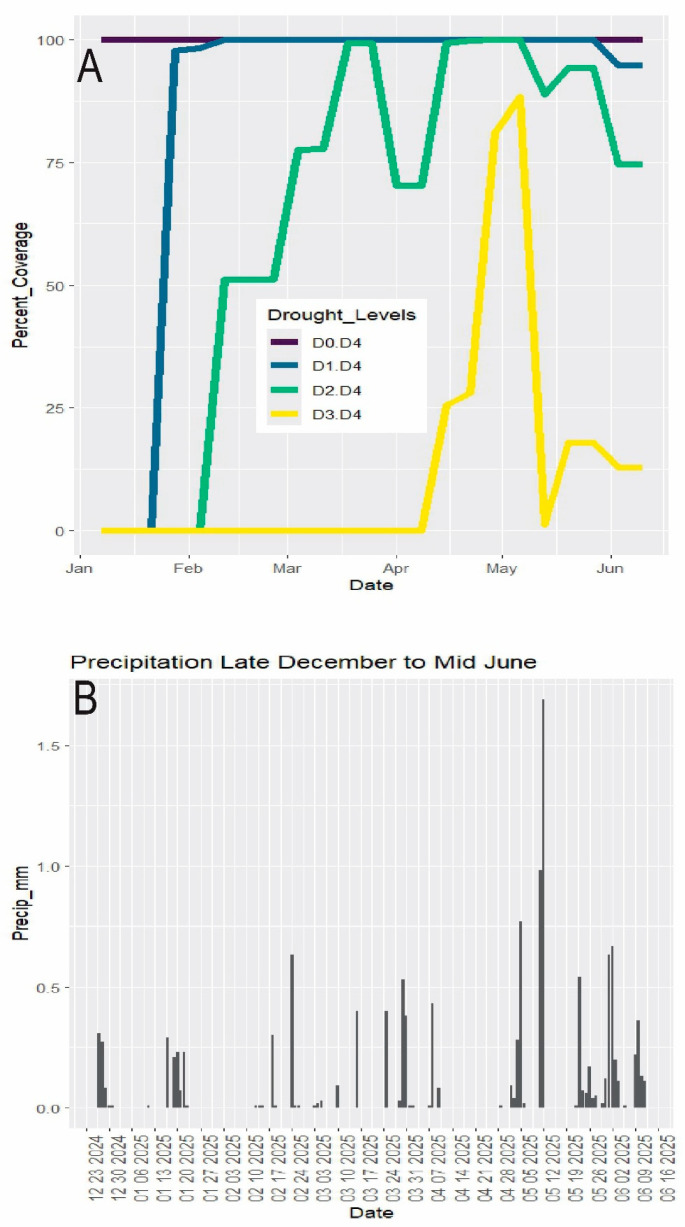
Weather data. Dry season December–mid June 2025. (**A**) Drought. D0 = abnormally dry. D1 = moderate drought. D2 = severe drought. D3 = extreme drought. D4 = exceptional drought. Data from U.S. Drought Monitor (2025). Percent_Coverage refers to the percent of Florida land area under each drought category. (**B**) Rainfall (data from SFWMD 2025).

**Figure 3 biology-14-01038-f003:**
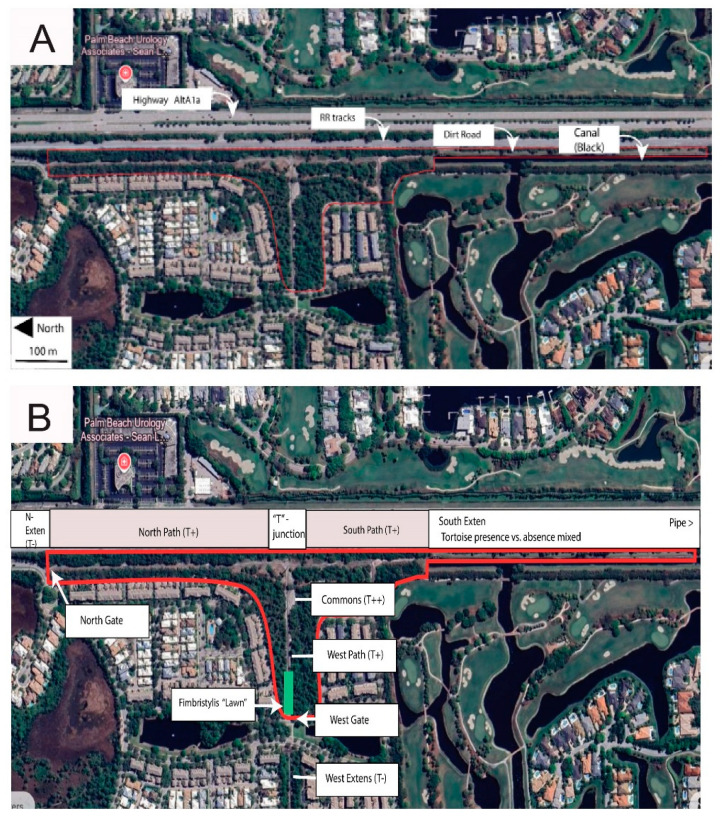
Study site (base photos from Google Earth, Airbus Imagery from the date 3/2/2025). (**A**) Major north-south axes. (**B**) Points of particular interest.

**Figure 4 biology-14-01038-f004:**
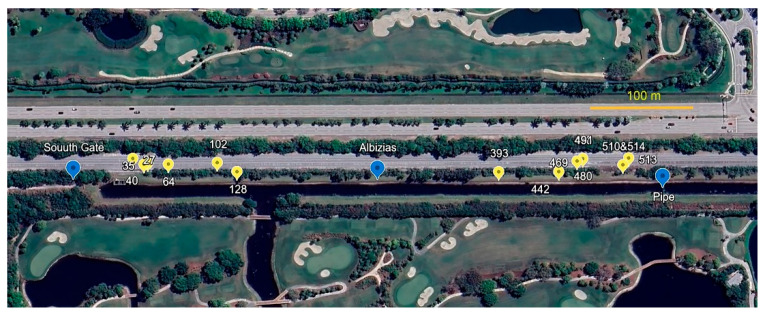
South extension. Yellow markers are burrows. Blue markers are geographic points mentioned in text (base photo from Google Earth, Airbus Imagery 3/2/2025).

**Figure 5 biology-14-01038-f005:**
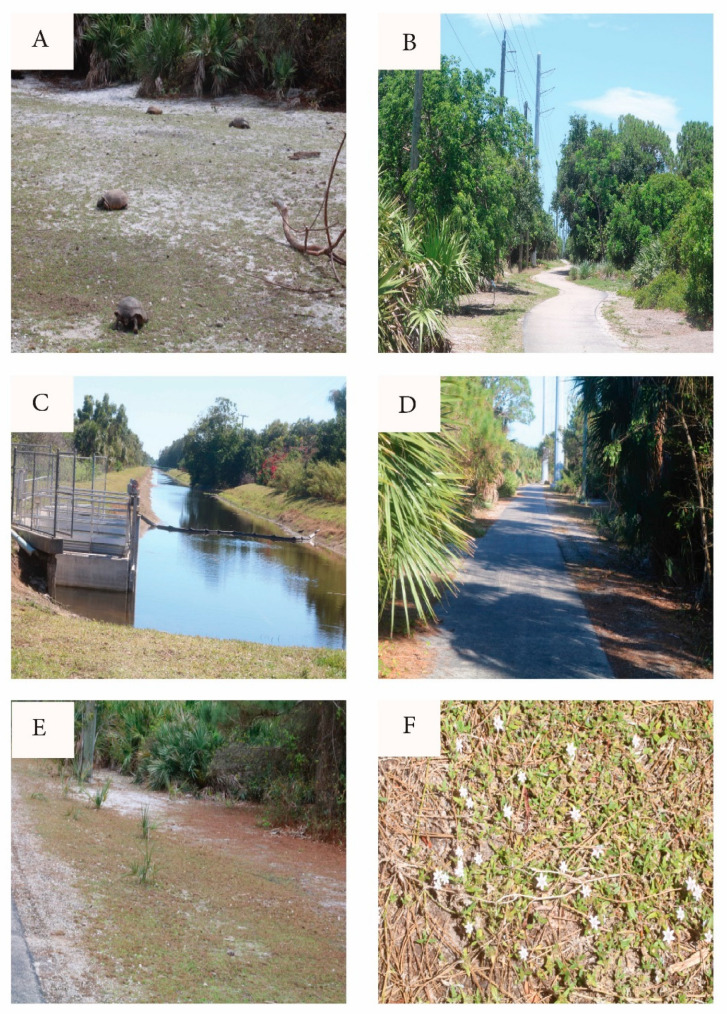
Study site views. (**A**) Commons with tortoises and *Richardia*-carpet. (**B**) North path looking north from commons. (**C**) South extension berm (left side) looking south from south gate. Right side is golf course. RR track is immediately to the left of the line of trees at left side of photo. (**D**) West path looking east toward commons. (**E**) *Fimbristylis* “lawn” along west path. (**F**) *Richardia*-carpet.

**Figure 6 biology-14-01038-f006:**
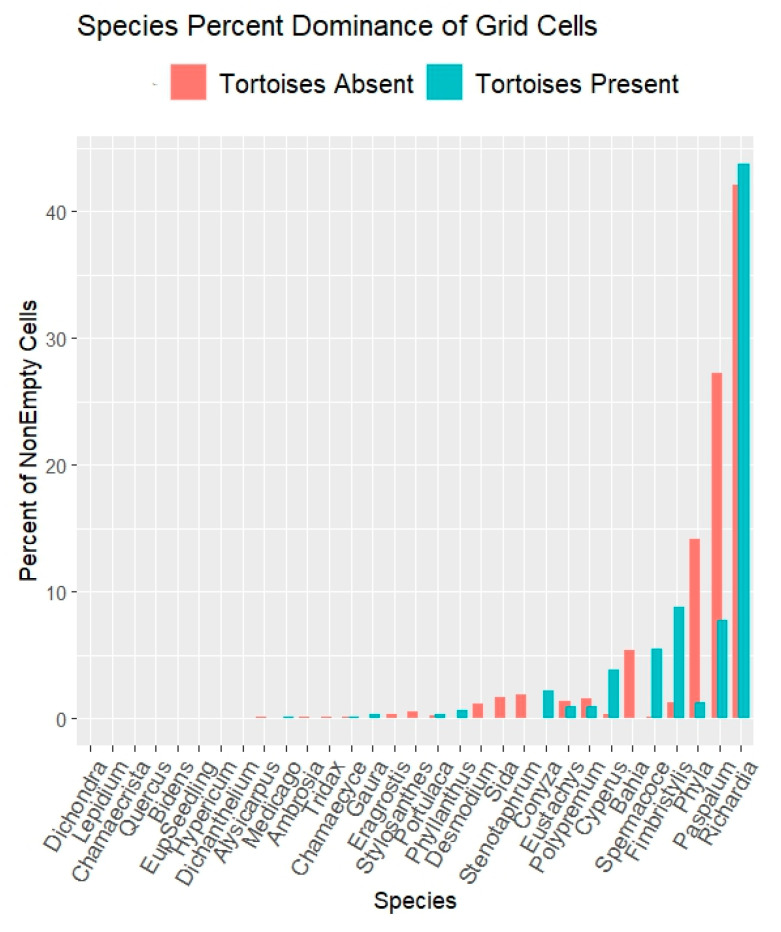
Grid quadrat results (64 cells/quadrat) on open carpet vegetation. Results show percents of grid cells dominated by each species. Full species names are in Appendix A.

**Figure 7 biology-14-01038-f007:**
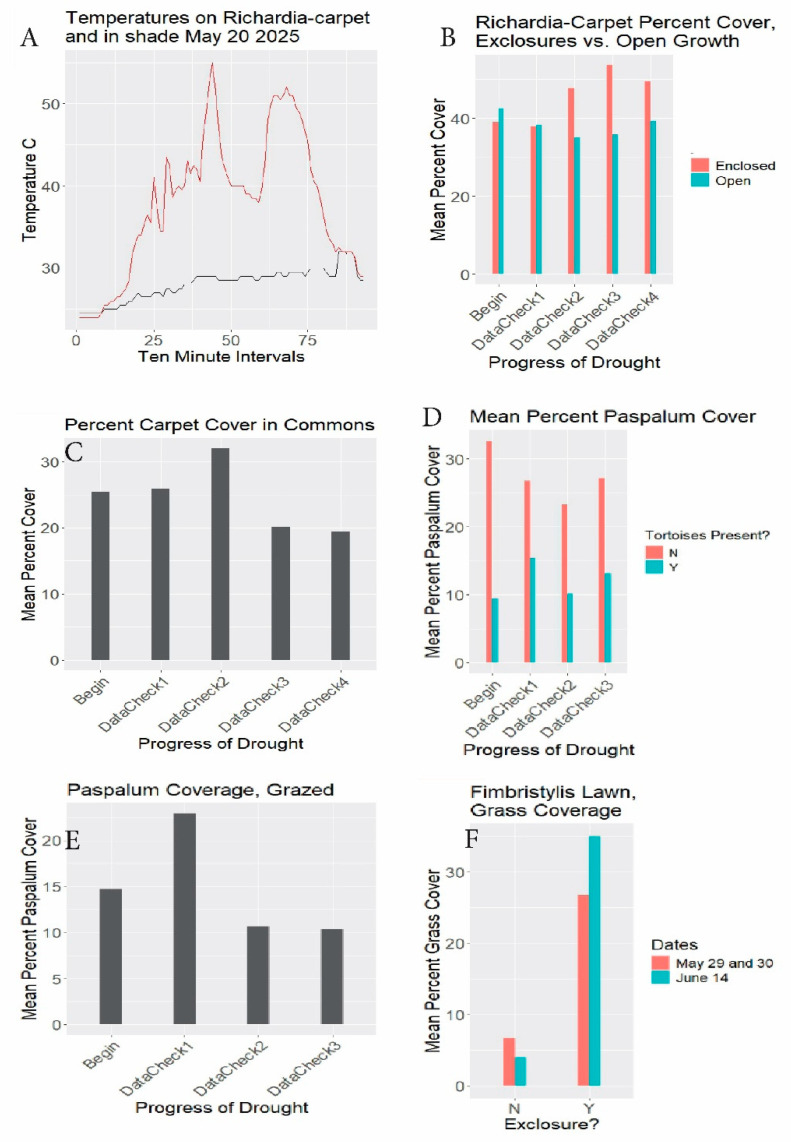
(**A**) Ground surface temperatures °C 5/20/2025 at south gate. Gold line: exposed ground surface. Dark line: adjacent shade. Horizontal axis: 10-minute intervals from 5:40 am DST. (**B**) *Richardia*-carpet exclosure results. Begin (setup): 3/3–3/19/2025. DataCheck1: 3/21–3/29, DataCheck2: 4/12–4/22, DataCheck3: 5/2–5/3, DataCheck4: 5/19–5/20. Percent cover = mean percents of quadrat areas covered with *Richardia*-carpet. DataChecks are periods during which readings from scattered quadrats were tallied. (**C**) Commons results. Begin: 3/7/2025, DataCheck1: 3/21, DataCheck2: 4/17, DataCheck3: 5/2, DataCheck4: 5/18-5/20. Percent cover = mean percents of quadrat areas covered with *Richardia*-carpet. (**D**) *Paspalum* 3030 photo results. Begin: 3/10/2025, DataCheck1: 4/10, DataCheck2: 5/5-5/6, DataCheck3: 5/21-5/22. “3030” refers to 30 rosette photos in area with tortoises present, and 30 in area with tortoises absent. Mean percent *Paspalum* coverage refers to percents of *Paspalum* rosettes centered in the quadrats covering the areas within the quadrats. Y = tortoises present (open to grazing). N = tortoises absent. (**E**) *Paspalum* transect results. Begin: 3/13–3/14/2025, DataCheck1: 4/6–4/7, DataCheck2: 5/5–5/7, DataCheck3: 5/21–5/23. Mean percent *Paspalum* coverage refers to percents of *Paspalum* rosettes covering the area within the quadrats. (**F**) *Fimbristylis* exclosure results. Begin: 4/1–4/2/2025, DataCheck1: 5/29–5/30 (brown bars), DataCheck2: 6/14 (blue bars). Percent grass coverage refers to percent of 31.5 cm × 22 cm grazed (left, N) and enclosed (right), Y) quadrats covered by varied grass species.

**Figure 8 biology-14-01038-f008:**
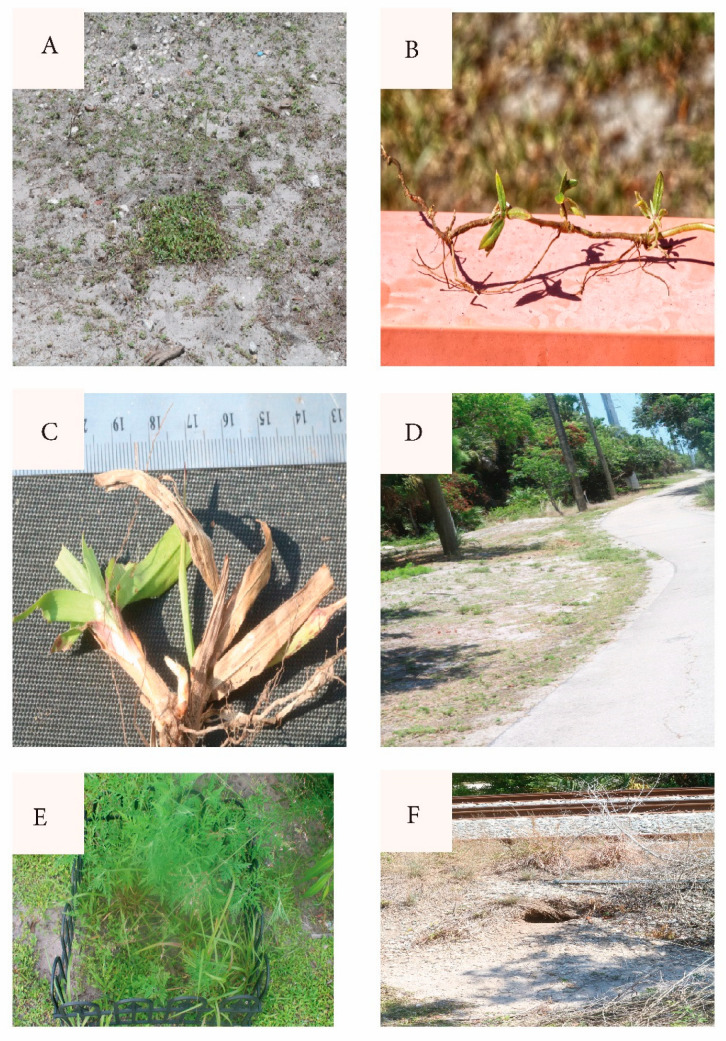
(**A**) *Richardia* exclosure (removed for photo) and surrounding grazed carpet along north path. June 16, 2025. (**B**) *Richardia* specimen from grazed south gate area, nodal rooting. (**C**) *Paspalum setaceum* grazed (note truncated leaves) and tillering (new shoot white, left of basal center). (**D**) North path June 16, 2025. Note enhanced growth along pavement, presumably from water and dissolved nutrient collection. (**E**) Exclosure at south gate, with *Ambrosia artemisiifolia*. (**F**) Tortoise burrow into shoulder of RR bed.

**Table 1 biology-14-01038-t001:** Results summary. Net changes in percent coverages within quadrats from beginnings to ends of substudies. Dates are in figure captions. All except *Fimbristylis* span from early drought to approximate end of drought.

	Begin_Grazed	End_Grazed	Grazed during Drought, Net Change	Begin_Ungrazed	End_Ungrazed	Ungrazed through Drought, Net Change	Figures
*Richardia* exclosures and quadrats	42.5%	39.4%	7.3% decrease	39%	49.5%	27% increase	Figure 7B
Commons quadrats	25.4%	19.4%	23.6% decrease	NA	NA	NA	Figure 7C
*Paspalum* 3030	9.5%	13.1%	38% increase	32.7%	27.1%	17.1% decrease	Figure 7D
*Paspalum* transects	14.7%	10.4%	29.3% decrease	NA	NA	NA	Figure 7E
*Fimbristylis* lawn, Poaceae coverage (note *Fimbristylis* data cover only end of drought period and early rains)	6.7%	4%	40.3% decrease (in grass coverage)	26.8%	35%	30.6% increase (in grass coverage)	Figure 7F

## Data Availability

All data are in the Appendix A cited above.

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
