# Peer review of "Botanical Assessment of Disturbed Urban Population of Threatened Gopher Tortoise (Gopherus polyphemus) Habitat in SE Florida During Drought"

_biology, 2025, doi:10.3390/biology14081038_

Round 1

Reviewer 1 Report

Comments and Suggestions for Authors

The article unfolds a study of the adaptive-competitive relationships of key groundcover plants under the dual stresses of grazing and drought by field monitoring plant responses in terrapin habitat during the dry season in South Florida. The article is thorough, graphically rich, and logical. It is recommended that the language be more written and that key background information be added in some paragraphs to complete the argument.

Line 50: "the species has a history of overlap" is a broad description. Suggest adding explanations or examples.

Line 65: "year-round additional stresses" is too broad. Suggest simple descriptions or examples.

Line 193: "Natural tortoise immigration is almost impossible." The source of the information needs to be indicated as further proof (e.g., field trips, interviews, etc.).

Line 206: This paragraph covers a wide range of topics, including turtle mortality, other herbivores, human disturbance, fence removal, and vegetation structure. Despite the detailed information, the paragraph structure is slightly confusing and it is recommended that it be organised hierarchically according to structures such as ecological factors, management actions, and hierarchical organisation of plant descriptions.

Chapter 2.2.4: Chapter headings are overly concise and slightly ambiguous; appropriate expansion is recommended.

Author Response

  1. The article unfolds a study of the adaptive-competitive relationships of key groundcover plants under the dual stresses of grazing and drought by field monitoring plant responses in terrapin habitat during the dry season in South Florida. The article is thorough, graphically rich, and logical. It is recommended that the language be more written

Yes, thank you, I reviewed the article with that in mind, and rewrote portions to add information and clarity.

and that key background information be added in some paragraphs to complete the argument.

Yes, agreed and done.

Line 50: "the species has a history of overlap" is a broad description. Suggest adding explanations or examples.

Thank you, this was done.

Line 65: "year-round additional stresses" is too broad. Suggest simple descriptions or examples.

Good point, examples were added.

Line 193: "Natural tortoise immigration is almost impossible." The source of the information needs to be indicated as further proof (e.g., field trips, interviews, etc.).

That portion was reworded with explanation.

Line 206: This paragraph covers a wide range of topics, including turtle mortality, other herbivores, human disturbance, fence removal, and vegetation structure. Despite the detailed information, the paragraph structure is slightly confusing and it is recommended that it be organised hierarchically according to structures such as ecological factors, management actions, and hierarchical organisation of plant descriptions.

Right---this portion of the ms. was rewritten as suggested.

Chapter 2.2.4: Chapter headings are overly concise and slightly ambiguous; appropriate expansion is recommended.

This has occurred.

Reviewer 2 Report

Comments and Suggestions for Authors

The study "Botanical assessment of the disturbed urban population of the endangered Gopher Tortoise (Gopherus polyphemus) habitat in southeast Florida during drought" monitors the main herbaceous plants that constitute the food source of a precarious urban population in Palm Beach County, Florida, during the severe drought of late winter and 
spring 2025. The main conclusion was that the main nutritive plants, including an exotic weed (Mexican clover, Richardia grandiflora), demonstrated  substantial resistance to drought and
drought combined with intense grazing.
The paper is of interest because it focuses on the study of an endangered species in an urban habitat, whereas most studies focus on protected species in protected areas. The paper also demonstrates that urban areas could be transformed into favorable habitats for various species with minimal management efforts, and that these habitats, often reserved exclusively for humans, are or could be more friendly to species facing difficulties. 
The paper is clearly presented, the questions are clear, and the methodology is also presented in detail and clearly. I consider question 3 to be redundant with question 2, so one of the two can be eliminated.
The conclusions are supported by the research results and are also presented in detail, along with the limitations of this study and recommendations for future studies.

Author Response

  1. The study "Botanical assessment of the disturbed urban population of the endangered Gopher Tortoise (Gopherus polyphemus) habitat in southeast Florida during drought" monitors the main herbaceous plants that constitute the food source of a precarious urban population in Palm Beach County, Florida, during the severe drought of late winter and 
    spring 2025. The main conclusion was that the main nutritive plants, including an exotic weed (Mexican clover, Richardia grandiflora), demonstrated  substantial resistance to drought and
    drought combined with intense grazing.
    The paper is of interest because it focuses on the study of an endangered species in an urban habitat, whereas most studies focus on protected species in protected areas. The paper also demonstrates that urban areas could be transformed into favorable habitats for various species with minimal management efforts, and that these habitats, often reserved exclusively for humans, are or could be more friendly to species facing difficulties. 
    The paper is clearly presented, the questions are clear, and the methodology is also presented in detail and clearly. I consider question 3 to be redundant with question 2, so one of the two can be eliminated.

The change took place.

The conclusions are supported by the research results and are also presented in detail, along with the limitations of this study and recommendations for future studies.

Reviewer 3 Report

Comments and Suggestions for Authors

The paper deals with monitoring of weedy foodplant populations in relation to grazing by threatened gopher tortoise and drought in Southeastern Florida. The performance of alien plant populations in relation to various environmental factors is of global importance, particularly when dealing with invasive species. This is even more important as the threatened reptile species is involved in the study as a grazer. The study presents a lot of supplementary materials including videos. The paper is clearly structured, particularly, regarding the research questions, which facilitates its reading and perception of the whole study. In general, the manuscript looks well prepared for the publication. However, some minor points should be considered before publication, which are as follows:

– Figure 2 lacks some explanation, A: What is Percent_Coverage on vertical axis? What is D4? B: To be correct, this is “Precipitation Late December to Mid June”.

– Abbreviations, like SM (line 113), should be written in full words at the first mention.

– Line 132: “path runs from the commons southward approx. 258 mm to the freely transited…” – I suppose it runs 258 metres, not millimetres.

– Line 227: Is the term “subsground” correct?

– Line 255: Replace “marked 60 X 40 marked quadrat” with “marked 60 X 40 quadrat”.

– Line 333: The title of Table 1 needs clarification of what “Net changes” are meant. I would suppose it is species cover/abundance expressed in percentage of grid cells in a quadrat?

– Line 363: Replace “Table 2” with “Table 1”.

– Lines 373–374: Replace “mitored stresses” with “minored stresses”.

– Line 382: Replace “strile substrates” with “sterile substrates”.

– Line 411: Replace "drought-dromant” with “drought-dormant”.

It would be quite reasonable to move Figure 6 from Materials and Methods to Results as it says in the caption: “grid quadrat results (64 cells/quadrat) on open carpet vegetation”. And the reference to Fig. 6 is already given in Results (line 329). Similarly, Figure 7 should be a part of Results, not Materials and Methods.

Supplementary Materials F. Geographic points, Row 5: Check if the longitude in the coordinate “26 54 17             80 95 42” is okay.

Author Response

  1. The paper deals with monitoring of weedy foodplant populations in relation to grazing by threatened gopher tortoise and drought in Southeastern Florida. The performance of alien plant populations in relation to various environmental factors is of global importance, particularly when dealing with invasive species. This is even more important as the threatened reptile species is involved in the study as a grazer. The study presents a lot of supplementary materials including videos. The paper is clearly structured, particularly, regarding the research questions, which facilitates its reading and perception of the whole study. In general, the manuscript looks well prepared for the publication. However, some minor points should be considered before publication, which are as follows:

– Figure 2 lacks some explanation, A: What is Percent_Coverage on vertical axis? What is D4? B: To be correct, this is “Precipitation Late December to Mid June”.

These observations were all addressed as indicated, thank you, and as a consequence Fig2A was replaced with no alteration in content for better image quality.

– Abbreviations, like SM (line 113), should be written in full words at the first mention.

Has now occurred

– Line 132: “path runs from the commons southward approx. 258 mm to the freely transited…” – I suppose it runs 258 metres, not millimetres.

Fixed thanks

– Line 227: Is the term “subsground” correct?

Corrected to substrate

– Line 255: Replace “marked 60 X 40 marked quadrat” with “marked 60 X 40 quadrat”.

Done as suggested

– Line 333: The title of Table 1 needs clarification of what “Net changes” are meant. I would suppose it is species cover/abundance expressed in percentage of grid cells in a quadrat?

Yes thanks, this has now been clarified.

– Line 363: Replace “Table 2” with “Table 1”.

Fixed thank you

– Lines 373–374: Replace “mitored stresses” with “minored stresses”.

Fixed thank you

– Line 382: Replace “strile substrates” with “sterile substrates”.

Fixed thank you

– Line 411: Replace "drought-dromant” with “drought-dormant”.

Fixed thank you

It would be quite reasonable to move Figure 6 from Materials and Methods to Results as it says in the caption: “grid quadrat results (64 cells/quadrat) on open carpet vegetation”. And the reference to Fig. 6 is already given in Results (line 329). Similarly, Figure 7 should be a part of Results, not Materials and Methods.

Relocation has occurred.

Supplementary Materials F. Geographic points, Row 5: Check if the longitude in the coordinate “26 54 17             80 95 42” is okay.

Corrected to 05 42, thank you